# Concept design of hybrid-actuated lower limb exoskeleton to reduce the metabolic cost of walking with heavy loads

**Qiaoling Meng**[1,2,3], **Bolei Kong**[1,2,3], **Qingxin Zeng**[1,2,3], **Cuizhi Fei**[1,2,3], **Hongliu Yu**[1,2,3]*

**1** Institute of Rehabilitation Engineering and Technology, University of Shanghai for Science and Technology, Shanghai, China, **2** Shanghai Engineering Research Center of Assistive Devices, Shanghai, China, **3** Key Laboratory of Neural-Functional Information and Rehabilitation Engineering of the Ministry of Civil Affairs, Shanghai, China

* yhl_usst@outlook.com

**Data Availability Statement:** We performed this study using the OpenSim software package, which is open-source and freely available at https://opensim.stanford.edu. The muscle energetics

## Abstract

This paper proposes the conceptual design method for a hybrid-actuated lower limb exoskeleton based on energy consumption simulation. Firstly, the human-machine coupling model is established in OpenSim based on the proposed three passive assistance schemes. On this basis, the method of simulating muscle driving is used to find out the scheme that can reduce the metabolic rate the most with 3 passive springs models. Then, an active-passive cooperative control strategy is designed based on the finite state machine to coordinate the operation of the power mechanism and the passive energy storage structure and improve the mobility of the wearer. In the end, a simulation experiment based on the human-machine coupled model with the addition of active actuation is proceeded to evaluate its assistance performance according to reducing metabolic rate. The results show that the average metabolic cost decreased by 7.2% with both spring and motor. The combination of passive energy storage structures with active actuators to help the wearer overcome the additional consumption of energy storage can further reduce the body's metabolic rate. The proposed conceptual design method can also be utilized to implement the rapid design of a hybrid-actuated lower limb exoskeleton.

## Introduction

A lower limb exoskeleton robot is a kind of mobility auxiliary device for people with lower limb dysfunctions, such as a stroke, a spinal cord injury, or the loss of a limb [1]. Reducing the metabolic cost that is very important to the design of this kind of exoskeleton, especially for the ones utilized by the patients [2]. According to the research work of decades, the most common lower limb exoskeleton robot is designed as joints actuated directly. The wearers have to cost more energy on the exoskeletons because of their heavy weight. Such exoskeletons can be utilized in restoring function to patients who are without balance. But patients who can move by themselves could cost more energy when they are wearing this kind of exoskeleton. In this context, the kind of unpowered exoskeletons has been developed to improve the wearers' mobility. For instance, Jiang et al. designed an unpowered lower limb exoskeleton. The

model used in this study is available in OpenSim 4.3. The analyses performed in this study were based on simulations originally generated by Dembia and Delp (2017), which are freely available at https://simtk.org/projects/assistloadwalk.

**Funding:** All authors were gratefully supported by the National Natural Science Foundation of China (62073224). The funders had no role in study design, data collection and analysis, decision to publish, or preparation of the manuscript.

**Competing interests:** The authors have declared that no competing interests exist.

specially shaped torsion spring and cam cooperate to form an energy storage mechanism, which is placed at the hip joint to compensate for the weight of the thigh and help the hip joint flexion [3]. These types of exoskeletons often provide power from the stored energy by the spring elements during walking. Yet, such exoskeletons cannot be utilized by patients who can be hindered by them. Therefore, an external power should be designed in a novel lower-limb exoskeleton robot.

The hybrid-actuated lower limb exoskeleton has two parts, one is a passive energy storage structure, and the other one is a powered mechanism. Researchers have designed different passive energy storage structures for unpowered exoskeletons. Zhou et al. designed a wearable hip joint exoskeleton, using 3-D printing technology to create waist and thigh braces to adapt to the irregular surface of the human body, setting springs in front of the hip joint, to recover negative mechanical energy and release the stored energy to assist hip flexion [4]. Ben-David et al. designed a knee joint exoskeleton, the passive energy storage structure is designed with a spring in the front of the knee joint to store up energy when the knee flexing and release mechanical energy when the knee extends, increasing the height of vertical jumps [5]. Collins et al. proposed an unpowered exoskeleton that employs a lightweight frame to place springs on the back of the calf parallel to the Achilles tendon and engages the springs with a mechanical clutch to improve the wearers' mobility [6]. Most of these researches are focused on single-joint assisting. The energy storage structures are designed with springs. Even they studied the wearer's metabolic cost, for instance, Collins' exoskeleton can reduce the metabolic cost of walking by 7.26±2.6% [6]. It is still confusing to the researchers which position of the structure can effectively reduce human metabolism. In addition, storing energy is a process of energy transformation. The wearers need to cost more energy during the process of storing energy. Therefore, the powered mechanism is integrated into the unpowered exoskeleton. A new problem has arisen that the cooperation of the energy storage structure and the powered mechanism.

In the research of powered exoskeletons, the control strategy is used to realize the coordinated movement of the gait between the wearer and the lower limb exoskeleton robotic system. An excellent control strategy can provide appropriate assistance for the wearer during the gait process and greatly reduce metabolic consumption. Therefore, researchers have made many efforts in the design and optimization of control strategies. Kim et al. proposed an iterative IMU-based control algorithm for a portable soft exosuit for hip extension assistance. The control strategy is to detect maximal hip flexion by tracking the sign change of angular velocity from the IMU of the hip and to estimate the wearer's phase in the gait cycle based on gait events. This iterative controller always provides the right assist force to the hip joint by gradually adjusting the timing and magnitude of the motor position profile. The results have shown that the metabolic costs of walking and running were reduced by 12.2% and 8.2%, respectively [7]. For the problem that the optimal estimation period of metabolic cost is relatively long, Zhang et al. developed a control method called 'human-in-the-loop optimization' to rapidly estimate metabolic profiles and adjust ankle exoskeleton torque profiles to optimize the economics of walking and running after humans wear the exoskeleton [8]. To understand the effects of soft exosuit on human loaded walking, Ding et al. developed a reconfigurable multi-joint actuation platform that delivers synchronized forces to the ankle and hip joints and reduces the wearer's metabolic cost of walking by an average of 14.6% [9]. Therefore, it is key to propose a suitable control strategy for hybrid-driven lower-extremity exoskeletons. It can coordinate the movement of the human-machine system, and improve human mobility to reduce human metabolism. However, the kind of hyper-actuated lower-extremity exoskeletons is different from the exoskeletons mentioned above. It is very important to know how to control compliantly the exoskeleton with the active power and passive spring.

In addition, the researchers measured the auxiliary effect of the device through experiments and continuously optimized the device. But it is impractical to experimentally test all exoskeletons, and simulation can effectively solve this problem [10–12]. At present, most experiments use indirect calorimetry systems to measure whole-body oxygen consumption and carbon dioxide production and obtain the metabolic changes of subjects before and after exoskeleton wearing [13,14]. To ensure the accuracy of the test results, the subjects need to undergo adaptive training before the test, and each experiment needs to strictly control the state of the subjects before and after wearing the exoskeleton. It takes a lot of time and money to complete the entire experiment in this situation. Some researchers have used simulation to evaluate the effect of the exoskeleton system and achieved some results. Analyzing human walking assistance using experiment-based OpenSim simulations, Firouzi et al. demonstrate the advantages of the proposed design and control of BATEX, such as 9.4%reduction in metabolic cost during normal walking conditions [15]. Aiming at the problem of how to store/release gait energy with high efficiency for the conventional unpowered lower extremity exoskeletons, Wang et al. proposed an unpowered lower-limb exoskeleton. The stiffness and metabolic cost of relevant muscles in lower extremity joints are obtained based on OpenSim software, and the results provide a theoretical basis for the optimal design of unpowered lower extremity exoskeleton [16]. In the research, the measurement of the effect of the exoskeleton was realized using simulation, which avoided complicated experimental procedures and saved a lot of time and cost in the initial design of the exoskeleton. As a result, the contribution of this paper is to propose a novel concept design approach for a novel hybrid-actuated lower-limb exoskeleton. The optimal position and stiffness of the passive springs are also discussed in this paper in order to provide a reference for passive assist strategies for lower extremity weight-bearing walking.

The rest of this paper is organized as follows. In Section 2, three lower extremity exoskeleton assistance schemes are preset and mathematical models are established for these schemes. Based on the scheme in Section 2, musculoskeletal models under different assistance were established in Section 3, the metabolic rate changes under different assistance schemes were obtained by using OpenSim simulation, and the optimal scheme was obtained by comparison. In Section 4, an active and passive cooperative control strategy based on metabolic simulation is proposed by analyzing the metabolic rate changes, kinematics, and kinetic data. In the end, the effectiveness of the control strategy is verified by simulation in Section 5.

## Design and modeling of passive parts of hybrid-actuated lower limb exoskeletons

Taking the existing passive lower extremity exoskeletons as a reference, three auxiliary solutions for the passive lower extremity exoskeletons are pre-set in this paper, namely setting a tension spring on the front side of the hip joint [4], and setting a compression spring on the back side of the knee joint [17] and a tension spring on the back of the ankle joint [6]. The mathematical models of each auxiliary scheme are built in OpenSim with the musculoskeletal system. The springs of each scheme are arranged in a plane parallel to the sagittal plane, and the upper anchor point of each auxiliary scheme is named U, and the lower anchor point is named D. The straight-line distance between the upper and lower anchor points is as a reference when setting the initial length of the spring for different auxiliary schemes.

### Hip assist

The hip assistance model is established as shown in Fig 1. The spring, as the storing energy mechanism, is set in the front of the hip joint. The parameters of the model are presented as shown in Fig 1(a) and 1(b). The point $P_{hip}$ is defined as the hip rotation center. The upper

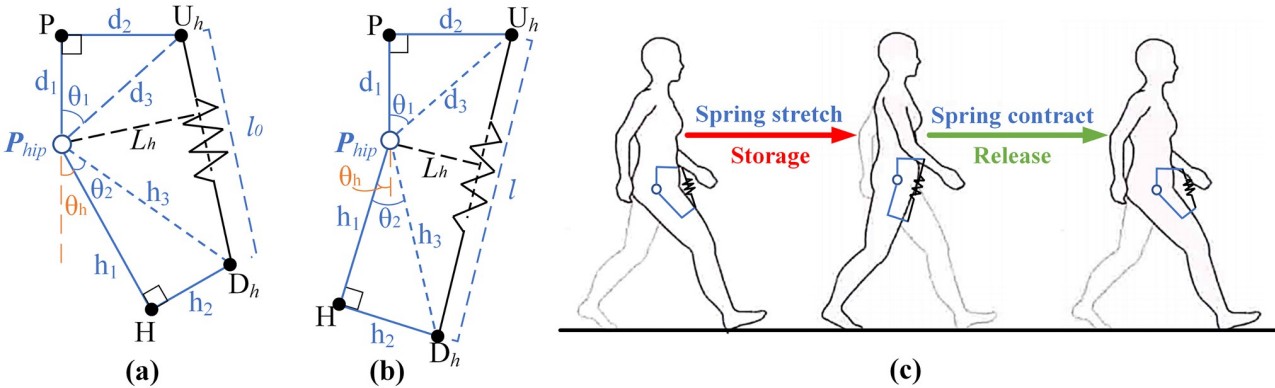

**Fig 1. Mathematical models of passive hip exoskeleton.** (a) The mathematical model of spring in initial state; (b) The mathematical model of spring in stretching state; (c) The layout of passive spring.

anchor point $U_h$ is set at the anterior superior iliac spine. The lower anchor point $D_h$ is set on the body surface of the front side of the thigh at half the length of femur. The surface $U_hD_hP_{hip}$ is located in the middle of the thigh and parallel to the sagittal plane of the human body. The spring is located between the anchor points $U_h$ and $D_h$, and its length is described as $l$ (the initial length is set as $l_0$). The point $P$ is the projection of point $U_h$ on the human torso, and the point $H$ is the projection of point $D_h$ on the central axis of the thigh. $d_1$ is the vertical distance from point $U_h$ to $P_{hip}$, and $d_2$ is the horizontal distance from point $U_h$ to $P_{hip}$. $h_1$ is the distance from point $H$ to $P_{hip}$, and $h_2$ represents the distance between point $D_h$ and $H$. The orange dotted line in Fig 1(a) and 1(b) is the extension line of $PP_{hip}$, it is defined as zero position. The angle between the orange dotted line and $P_{hip}H$ is the hip angle described as $\theta_h$ (Fig 1(a) shows hip flexion with positive angles, Fig 1(b) shows hip extension with negative angle). The work process of the spring is presented in Fig 1(c) in the human walking procedure. The spring is stretched to store energy during the support phase, and then the spring releases energy to assist in flexion of the hip joint during the swing phase.

Therefore, the distance $d_3$ from the upper anchor point $U_h$ to the $P_{hip}$ is:

$$d_3 = \sqrt{d_1{}^2 + d_2{}^2} \tag{1}$$

The distance $h_3$ from the front thigh anchor point $D_h$ to the $P_{hip}$ is:

$$h_3 = \sqrt{h_1{}^2 + h_2{}^2} \tag{2}$$

In this paper, the initial length of the spring $l_0$ is set the initial length at the position of the maximum flexion angle $\theta_h$ of the hip joint to increase the capability of storing energy. Therefore, the included angle $\theta_1$ between $d_1$ and $d_3$, and the included angle $\theta_2$ between $h_1$ and $h_3$ can be obtained.

$$\theta_1 = arc \tan \frac{d2}{d1} \tag{3}$$

$$\theta_2 = arc \tan \frac{h2}{h1} \tag{4}$$

The relationship between spring length $l$ and the hip joint angle $\theta_h$ can be obtained as

follows.

$$l(\theta_h) = \sqrt{d_3^{\,2} + h_3^{\,2} - 2d_3 h_3 cos(\pi - (\theta_1 + \theta_2 + \theta_h))}, l \geq l0 \tag{5}$$

The spring stiffness is expressed as $K_h$, the functional relationship between spring force $F_h$ and hip angle $\theta_h$ can be obtained as follows.

$$F_h(\theta_h) = K_h \times (l(\theta_h) - l_0) \tag{6}$$

The vertical distance from $P_{hip}$ to $l$ is the moment arm of the moment generated by the spring at the hip joint. The moment arm $L_h(\theta_h)$ is:

$$L_h(\theta_h) = \frac{d_3 h_3 \sin(\pi - (\theta_1 + \theta_2 + \theta_h))}{l(\theta_h)} \tag{7}$$

The spring moment $M_h(\theta_h)$ is decided according to Eqs (6) and (7).

$$M_h(\theta_h) = F(\theta_h) \times L_h(\theta_h) \tag{8}$$

### Knee assist

The knee assistance model is established as shown in Fig 2. The spring, as the shorting energy mechanism, is set in the front of the hip joint. The parameters of the model are presented as

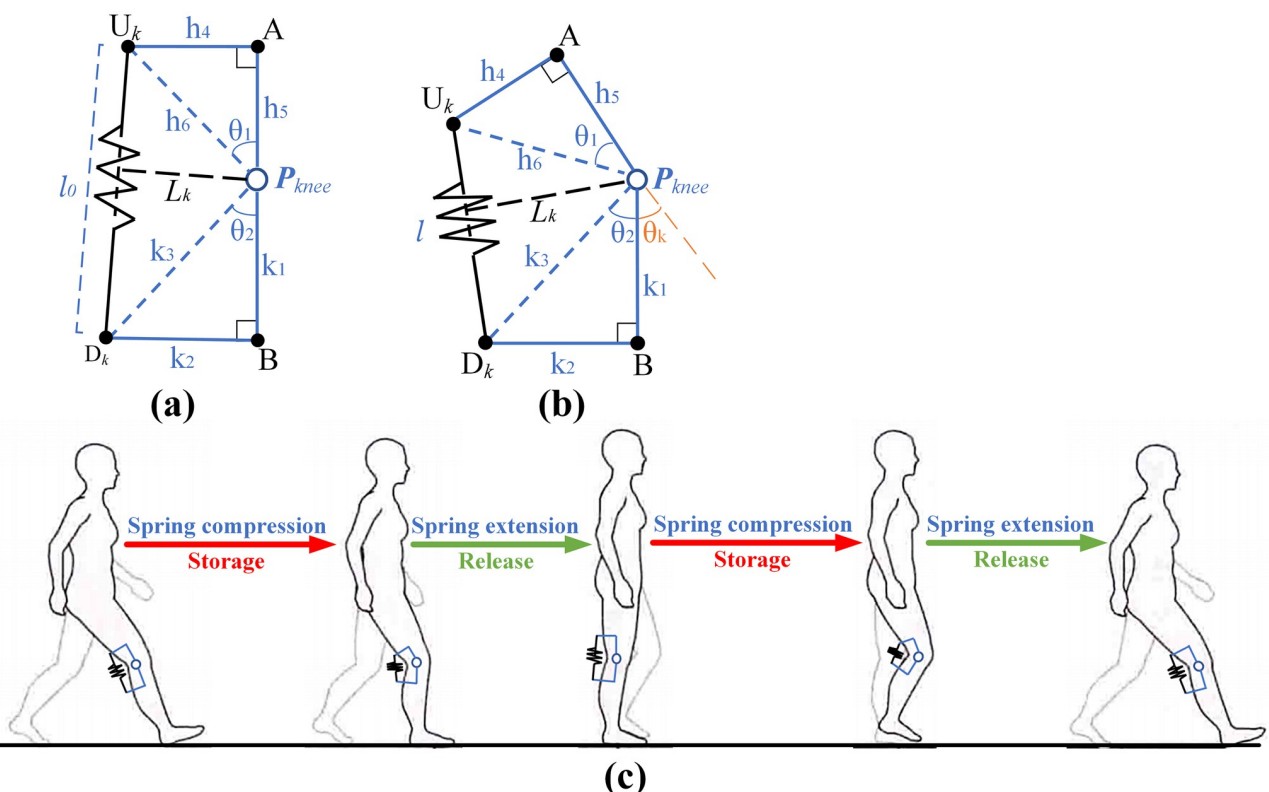

**Fig 2. Mathematical models of passive knee exoskeleton.** (a) The mathematical model of spring in initial state; (b) The mathematical model of spring in compressing state; (c) The layout of passive spring.

shown in Fig 2(a) and 2(b). The point $P_{knee}$ is defined as the knee joint rotation center. The upper anchor point $U_k$ is set at about 15cm above the knee joint on the back surface of the thigh. The lower anchor point $D_k$ is set at about 10cm below the knee joint on the back surface of the calf. The plane $U_k D_k P_{knee}$ is located in the middle of the calf and parallel to the sagittal plane of the human body. The spring is located between the anchor points $U_k$ and $D_k$, and the length is $l$ (the initial length is set to l0). The point $A$ is the projection of point $U_k$ on the central axis of thigh, and the point $B$ is the projection of point $D_k$ on the central axis of the calf. $h_4$ is the horizontal distance from point $U_k$ to $P_{knee}$, and $h_5$ is the vertical distance from point $U_k$ to $P_{knee}$. $k_1$ is the distance from point $B$ to $P_{knee}$, and $k_2$ is the distance between thepoint $D_k$ and $B$. The orange dotted line in Fig 2(b) is the extension line of $AP_{knee}$. The angle between the orange dotted line and $AP_{knee}$ is the knee angle $\theta_k$ (Fig 2(a) shows the state with the knee angle at 0, and Fig 2(b) shows knee flexion with always positive angle). The work process of the spring is presented in Fig 2(c) in the human walking procedure. The gait cycle begins with the knee joint in maximum extension and the compression spring at the initial length. Then the spring on the support leg is compressed to store energy, and when the opposite toe is off the ground, the spring releases the energy to help raise the body's center of gravity.

Therefore, the distance $h_6$ from the upper anchor point $U_k$ to the $P_{knee}$ is:

$$h_6 = \sqrt{h_4^2 + h_5^2} \tag{9}$$

The distance $k_3$ from the lower anchor point $D_k$ to the $P_{knee}$ is:

$$k_3 = \sqrt{k_1^2 + k_2^2} \tag{10}$$

In this paper, the initial length of the spring $l_0$ is set the initial length at the position of the minimum knee joint angle $\theta_k$ of the knee joint to increase the capability of storing energy. The included angle $\theta_1$ between $h_5$ and $h_6$, and the included angle $\theta_2$ between $k_1$ and $k_3$ can be obtained.

$$\theta_1 = arc \tan \frac{h_4}{h_5} \tag{11}$$

$$\theta_2 = arc \tan \frac{k_2}{k_1} \tag{12}$$

The relationship between spring length $l$ and the knee joint angle $\theta_k$ can be obtained as follows.

$$l(\theta_k) = \sqrt{h_6^2 + k_3^2 - 2h_6 k_3 \cos(\pi - \theta_1 - \theta_2 - \theta_k)}, l \leq l0 \tag{13}$$

The spring stiffness is expressed as $K_k$, the functional relationship between spring force $F_k$ and knee angle $\theta_k$ can be obtained as follows.

$$F_k(\theta_k) = K_k \times (l(\theta_k) - l_0) \tag{14}$$

The vertical distance from $P_{knee}$ to $l$ is the moment arm of the moment generated by the spring at the knee joint. The moment arm $L_k(\theta_k)$ is:

$$L_k(\theta_k) = \frac{h_6 k_3 \sin(\pi - \theta_1 - \theta_2 - \theta_k)}{l(\theta_k)} \tag{15}$$

The spring moment $M_k(\theta_k)$ is decided according to Eqs (14) and (15).

$$M_k(\theta_k) = F(\theta_k) \times L_h(\theta_k) \tag{16}$$

## Ankle assist

The hip assistance model is established as shown in Fig 3. The spring, as the storing energy mechanism, is set in the front of the ankle joint. The parameters of the model are presented as shown in Fig 3(a) and 3(b). The point $P_{ankle}$ is defined as the ankle joint rotation center. The upper anchor point $U_a$ is set at about 30cm above the ankle joint on the back surface of the calf. The lower anchor point $D_a$ is set at about 12cm behind the ankle joint. A surface located in the middle of the calf and parallel to the sagittal plane of the human body is formed by points $U_a$, $D_a$ and $P_{ankle}$. The spring is located between the anchor points $U_a$ and $D_a$, and the length is $l$ (the initial length is set as $l_0$). The point $C$ is the projection of point $U_a$ on the central axis of calf. a1 is the distance from point $C$ to $P_{ankle}$, and $a_2$ is the distance from point $U_a$ to $C$. The orange dotted line in Fig 3(a) and 3(b) is the vertical line of $a_1$. The angle between the orange dotted line and $a_4$ is the ankle angle $\theta_a$ (Fig 3(a) shows ankle plantar flexion with positive angles, Fig 3(b) shows ankle dorsiflexion with negative angle). The work process of the spring is presented in Fig 3(c) in the human walking procedure. The ankle joint is in maximal plantar flexion at the beginning of the gait cycle and the spring is in its initial state. The spring is stretched to store energy during the support period. The spring releases energy to help the body move forward during the pre-swing phase. The toe of the swing leg is off the ground, and

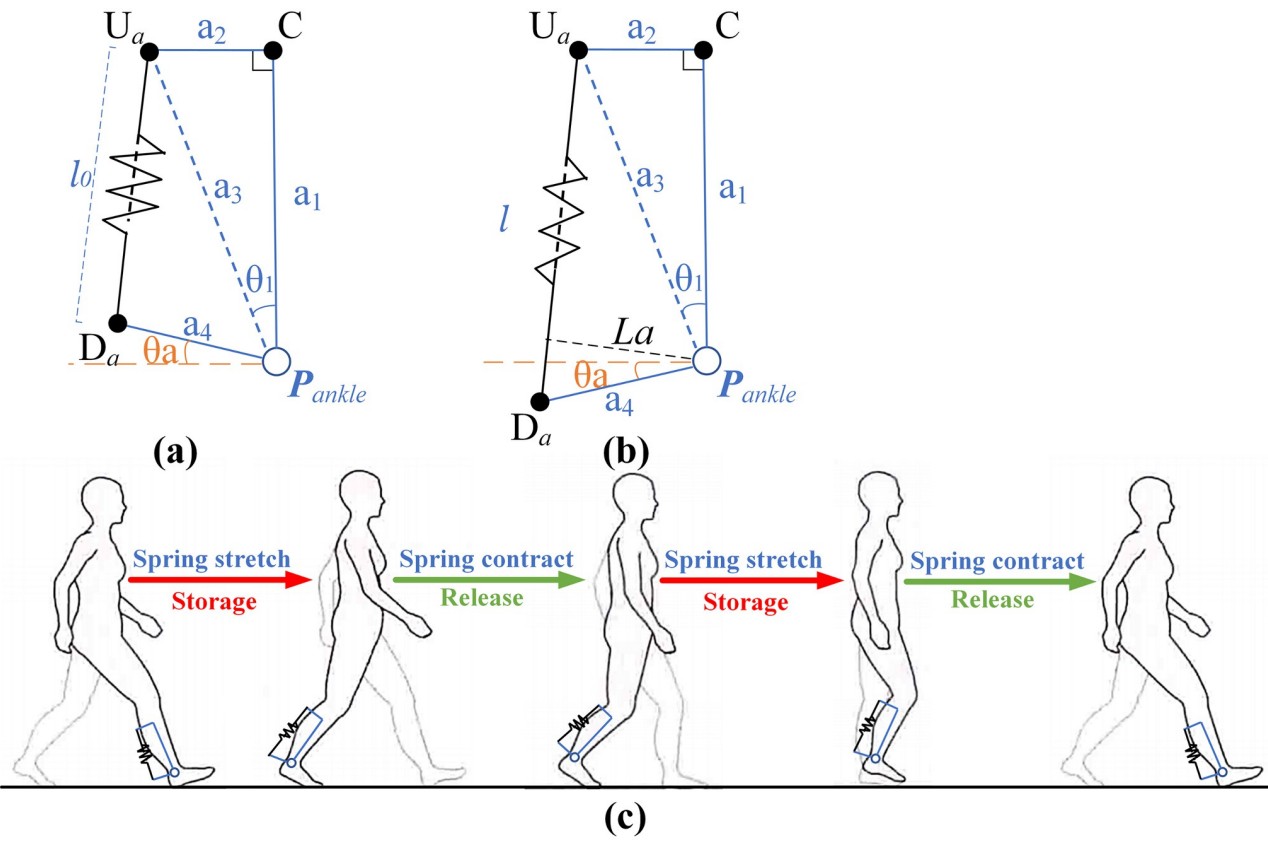

**(a)**   **(b)**

**(c)**

**Fig 3. Mathematical models of passive ankle exoskeleton.** (a) The mathematical model of spring in initial state; (b) The mathematical model of spring in stretching state; (c) The layout of passive spring.

the spring returns to its original length. During the swing phase, the ankle is dorsiflexed to prevent the toes from touching the ground, and the spring is slightly elongated to release energy for the next state cycle.

Therefore, the distance $a_3$ from the upper anchor point $U_a$ to $P_{ankle}$ is:

$$a_3 = \sqrt{a_1^2 + a_2^2} \tag{17}$$

In this paper, the initial length $l_0$ of the spring is set at the maximum plantar flexion angle $\theta_a$ of the ankle joint to increase the capability of storing energy. Therefore, the included angle $\theta_1$ between $a_1$ and $a_3$ can be obtained.

$$\theta_1 = arc \tan \frac{a_2}{a_1} \tag{18}$$

The relationship between spring length $l$ and the ankle joint angle $\theta_a$ can be obtained as follows.

$$l(\theta_a) = \sqrt{a_3^2 + a_4^2 - 2a_3a_4 \times cos\left(\frac{\pi}{2} + \theta_a - \theta_1\right)} \tag{19}$$

The spring stiffness is expressed as $K_a$, the functional relationship between spring force $F_a$ and hip angle $\theta_a$ can be obtained as follows.

$$F_a(\theta_a) = K_a \times (l(\theta_a) - l_0) \tag{20}$$

The vertical distance from $P_{ankle}$ to $l$ is the moment arm of the moment generated by the spring at the ankle joint. The moment arm $L_a(\theta_a)$ is:

$$L_a(\theta_a) = \frac{a_3a_4 \sin\left(\frac{\pi}{2} + \theta_a - \theta_1\right)}{l(\theta_a)} \tag{21}$$

The spring moment $M_a(\theta_a)$ is decided according to Eqs (20) and (21).

$$M_a(\theta_a) = F(\theta_a) \times L_a(\theta_a) \tag{22}$$

## Simulation based energy analysis

### Experimental data

In this study, we generated simulations of 7 male subjects using the data, models, and methods reported by Dembia and Delp [18]. The Stanford University Institutional Review Board approved the experimental protocol and subjects provided informed written consent. The data contains motion capture data and ground reaction force data for 7 subjects in their natural state and under weight-bearing conditions. The simulations in this paper have been proceeded in ideal state that massless with no torque or power limits. It is worth noting that there are many missing ground reaction force data from walking in the natural state in this dataset, so only the data measured in the load-bearing state is simulated and analyzed.

### Simulation of experiment

A 3D musculoskeletal model was used in OpenSim, as shown in Fig 4(a), which consists of 80 tendon units and 37 degrees of freedom [19]. Following the method of Dembia et al., the model was scaled by marking point data to match the body parameters of different subjects. In addition, a 38kg load was added to the human torso to meet the data of 7 subjects walking with

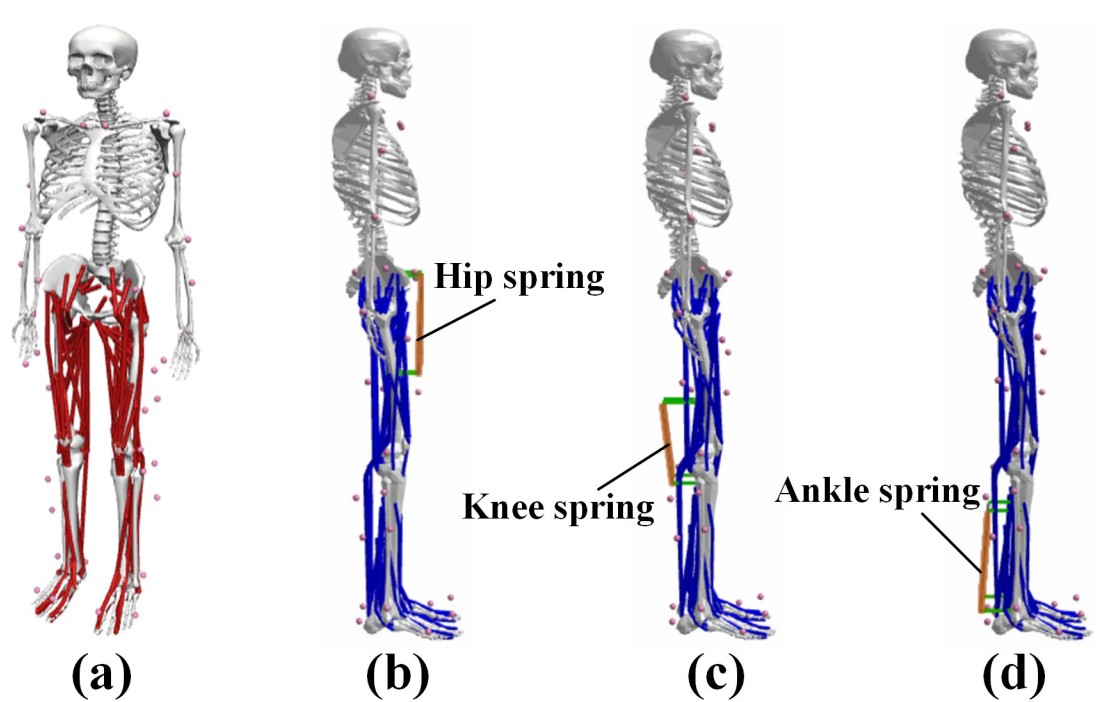

**Fig 4. The simulation models.** (a) The musculoskeletal model; (b) The hip joint exoskeleton model; (c) The knee joint exoskeleton model; (d) The ankle joint exoskeleton model.

a heavy load [18]. According to the three preset auxiliary schemes, springs are added to the corresponding joints of each adjusted model.

Based on the PointToPointSpring plug-in in OpenSim 4.3, spring actuators are created in each auxiliary scheme. This spring actuator is an ideal massless and dissipative spring. As shown in Fig 4(b)–4(d), the springs are added to the different joints of the mode. Set the upper and lower anchor points of the spring according to the mathematical model established in the second section. The initial length and stiffness of the spring need to be set before each simulation. Since the body parameters of different subjects are different, the initial length of the auxiliary spring added by each subject needs to be determined according to the body parameters and model.

As a standard component of passive exoskeletons, the energy storage springs have a fixed stiffness. The selection of the stiffness parameters of the exoskeleton energy storage spring is an important link in the research process. A high-stiffness spring will increase the discomfort of wearing, and insufficient stiffness will make it difficult to achieve proper walking effect [2]. Ten springs with different stiffness were selected for different assistance schemes, based on the ratio of the peak torque that the spring can produce to the peak joint torque at the auxiliary joint. The ratios are evenly distributed between 0 and 1 for hip and knee assist protocols. In the ankle joint exoskeleton model, the ratio is evenly distributed between 0 and 0.5. Because when the ratio is higher than 0.5, the torque output of the standby actuator will be too large (greater than 50NM), which will lead to inaccurate results in calculating muscle control [20].

Subsequently, the computed muscle control (CMC) tools in OpenSim is used to simulate muscle-driven movements [21]. To estimate the instantaneous metabolic rate of each subject under different auxiliary regimens from the simulations, the muscle metabolism model developed by Umberger [22,23] and further refined by Dembia and Delp [18] was used in this

study. Thus, we performed a total of 630 simulations (7 subjects, 3 assist regimens, 10 different stiffness springs, 3 walks in each condition).

## Simulation results

The simulated results of each subject's three walks under different assistance regimens and different stiffness conditions were averaged to obtain the average whole-body metabolic rate of each subject's weight-bearing walking under different assistance regimens and different stiffness conditions. Table 1 shows the reduction in overall mean whole-body metabolic rate for 7 subjects under the three auxiliary regimens. The first column is the subject's serial number and the height and weight of each subject. Subsequent columns correspond to the maximum mean metabolic rate reduction ratios for the subject's three auxiliary regimens. The symbol '-' in the table indicates that the metabolism has not decreased.

As can be seen from the simulation results in Table 1, the ankle-assisted regimen reduced the average metabolic rate of all subjects with the greatest magnitude. The hip and knee-assist regimen reduced metabolic rate in only some subjects, and the reduction was smaller. This is related to the weight bearing conditions of the subjects. The ankle joint assistance was chosen as a passive assistance scheme for hybrid-actuated lower extremity exoskeletons in this study to discuss the next control strategy design.

The whole-body metabolic rate was normalized to body weight to remove the effect of body weight differences as shown in Fig 5. Fig 5(a) shows the average whole-body metabolic rate under the passive ankle-assist regimen of the seven subjects. Fig 5(b) describes the results of the average metabolic rate under ankle spring assistance at different ratios.

As shown in Fig 5(a), the metabolic rise zone is approximately about 20~40% and 70~90%, and the average metabolic rate of walking with ankle assistance is higher than that of unassisted walking. It can be concluded that in the metabolic ascending region, the energy storage efficiency of the auxiliary springs on both sides of the ankle joint is greater than the energy release efficiency. If the motor can exert torque on the ankle joint in the metabolic rising area to compensate for the extra spring work, the metabolic rate can be further reduced. Therefore, a new control strategy for the hybrid-actuated exoskeleton can be proposed based on the metabolic cost results.

In the Fig 5(b), the red histogram is the average metabolic rate without the assist spring, and the rest are the average metabolic rate with 10 spring assists. When the ratio is greater than 0.35, increasing the spring rate will only increase the torque of the backup actuator. The data after 0.35 is distorted and is not used as a reference. The red line is the fitting curve of the average metabolic rate under the assistance of different stiffness springs. The ratio with the greatest decrease in metabolic rate was between 0.2 and 0.25.

**Table 1. Simulation results.**

| Subject (Height & Weight) | Assist joint | | |
|---|---|---|---|
| | Hip | Knee | Ankle |
| 01(1.92m 112kg) | - | 0.2% | 10.5% |
| 02(1.88m 89kg) | 1.2% | - | 3.3% |
| 03(1.91m 87kg) | - | - | 3.24% |
| 04(1.8m 64kg) | 0.45% | 0.8% | 11.6% |
| 05(1.83m 85kg) | 0.1% | 0.2% | 5.5% |
| 06(1.83m 67kg) | - | - | 9.6% |
| 07(1.83m 84kg) | - | 0.65% | 12.8% |

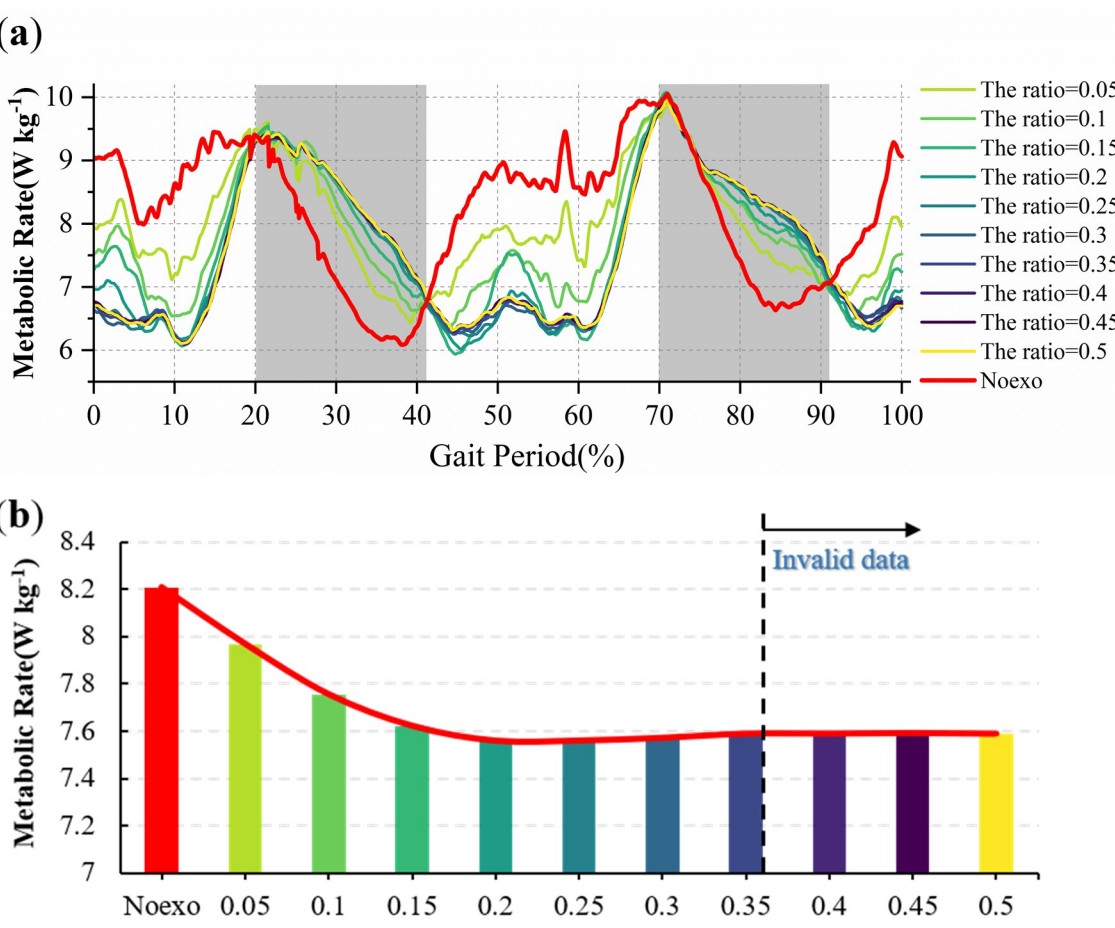

**Fig 5. Simulation results.** (a) The average whole-body metabolic rate with passive ankle exoskeleton assistance. The red line is the unassisted metabolic rate curve, and the rest of the color lines are the metabolic rate curves under the assistance of springs with different stiffnesses. The gray area is the metabolic increase area after adding the auxiliary spring. The white area is the metabolic decay area after adding an auxiliary spring. (b) The results of the average metabolic rate under ankle spring assistance at different ratios.

## Design of active and passive cooperative control strategy

The hybrid-actuated lower extremity exoskeletons combine power mechanisms with passive energy storage structures. The power mechanism needs to cooperate with the operation of the passive energy storage structure. It is very important to know how to cooperatively control the hybrid-actuated exoskeleton proposed in this paper. There are a lot of control strategies to control the complex exoskeleton. Finite State Machine (FSM) is one of the suitable control algorithms in the lower limb exoskeletons and prosthetics due to its rapid response. Such as Wang et al. [24], Ma et al. [25]. Therefore, a novel FSM-based control strategy for the hybrid-actuated exoskeleton is proposed according to Section 3.

### Finite states

Metabolic rate curves under the ankle-assisted strategy and lower extremity joint angles, angular velocities, and plantar pressures of seven subjects shown in Fig 6 are used to classify states in the gait cycle. It is worth noting that with the load increase, the proportion of the double-support period in the gait cycle during walking increases [26]. First, based on the mean ground

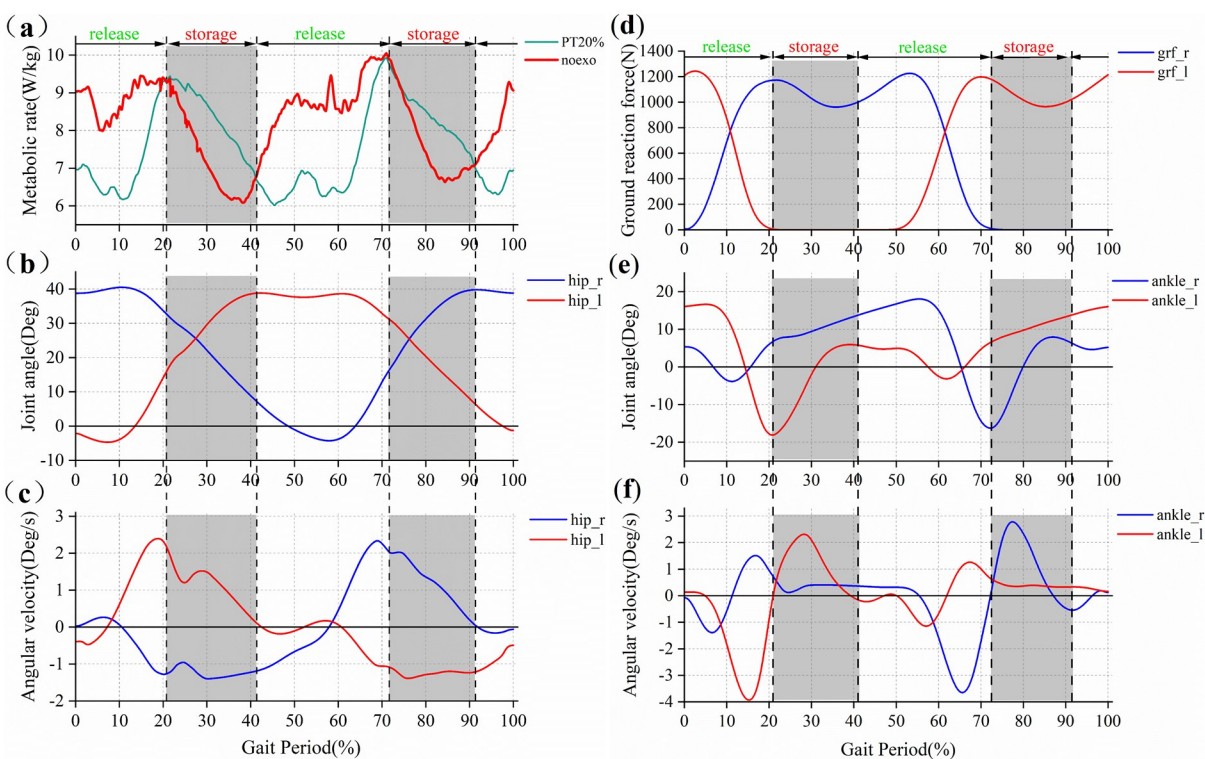

**Fig 6. Parameter basis for dividing states.** (a) The metabolic rate; (b) The hip joint angle; (c) The angular velocity, (d) The ground reaction force; (e) The ankle joint angle; (f) The angular velocity.

reaction force data of the subjects (Fig 6(d)), the double support phase (approximately 0~20% and 50~70% of the gait cycle) and the swing phase (approximately 20~50% and 70~100% of the gait cycle) are divided. Then, based on metabolic rate changes (Fig 6(a)), the metabolic rising phase (approximately 20~40% and 70~90% of the gait cycle) and the metabolic decline phase (approximately 0~20%, 40~70% and 90~100% of the gait cycle) are divided. Finally, the two division methods are combined to obtain six states of the gait cycle of weight-bearing walking, as shown in Fig 7, which are the right double support metabolic rate decline period (about 0~20%), the left swing metabolic rate rising period (about 20~40%), left swing

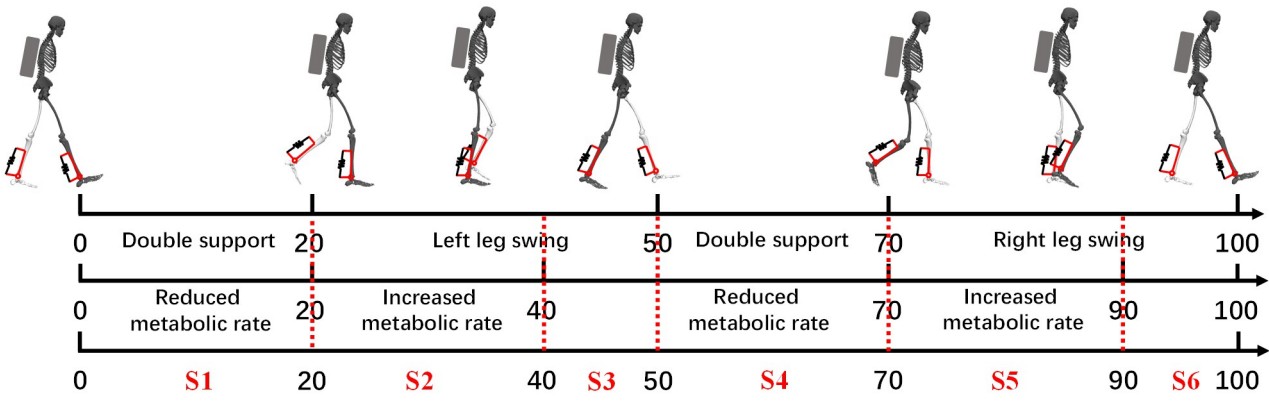

**Fig 7. Gait cycle division of wearing a passive ankle exoskeleton under load.**

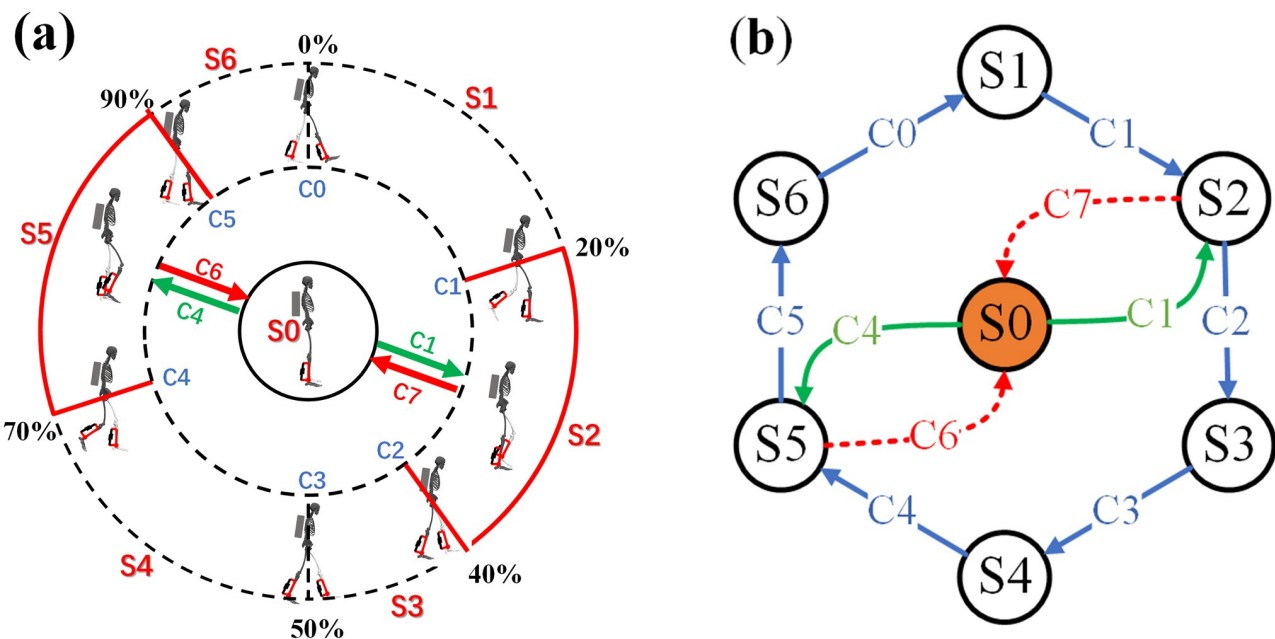

**Fig 8. Finite State Machine.** (a) Finite State Partitioning for Weighted Walking; (b)Finite State Machine Model.

metabolic rate decline period (about 40~50%), left double support metabolic rate decline period (about 50~70%), right swing metabolic rate rise period (about 70~90%) and right swing metabolic rate decline period (about 90~100%).

In addition, the stopping and restarting of the person during walking are also considered. As shown in Fig 8, the stopped state is set to S0. In the process of walking, there are two situations for stopping: the left leg is in front, the right leg is retracted when it stops, or the left leg is retracted when the right leg is stopped in front. To start walking, there are two situations: lift the left leg or lift the right leg. During the left swing period, the right leg is retracted when stopping and the left leg is raised when starting to walk. The retraction of the left leg at the stop of walking and the right leg lift at the beginning of walking both occur during the right swing phase. The specific parameters of the 7 states are as follows:

S0: Stop and stand in the swing period during walking, and when standing, both feet land ($P_r > 0$, $P_l > 0$), and the angular velocity of each joint should be zero ($\omega_{hr} \approx \omega_{hl} \approx \omega_{ar} \approx \omega_{al} \approx 0$).

S1: During this phase, the metabolic rate is lower than when walking without assistance, and the motor does not work. This state is the right double support period ($P_r > 0$, $P_l > 0$), and the right leg is in front of the left leg ($\theta_{hr} > \theta_{hl}$).

S2: During this phase, the metabolic rate is higher than that of walking without assistance, and the motor works. In this state, the right leg supports the swing of the left leg ($P_r > 0$, $P_l = 0$), and the hip angular velocity of the swinging leg is positive ($\omega_{hr} < 0$, $\omega_{hl} > 0$).

S3: During this phase, the metabolic rate is lower than when walking without assistance, and the motor does not work. In this state, the right leg supports the swing of the left leg ($P_r > 0$, $P_l = 0$), and the hip angular velocity of the swinging leg is negative ($\omega_{hr} < 0$, $\omega_{hl} < 0$).

S4: During this phase, the metabolic rate is lower than when walking without assistance, and the motor does not work. This state is the left double support period ($P_r > 0$, $P_l > 0$), and the left leg is in front and the right leg is behind ($\theta_{hr} < \theta_{hl}$).

**Table 2. The meaning of the symbol parameters used in the text.**

| Symbolic Parameters | Meaning |
|---|---|
| $P_r$, $P_l$ | Plantar pressure of right and left feet |
| $\theta_{hr}$, $\theta_{hl}$ | Right and left hip angle |
| $\omega_{hr}$, $\omega_{hl}$ | Right and left hip angular velocity |
| $\theta_{ar}$, $\theta_{al}$ | Right and left ankle angle |
| $\omega_{ar}$, $\omega_{al}$ | Right and left ankle angular velocity |
| $F_{sr}$, $F_{sl}$ | Right and left spring tension value |
| $M_{sr}$, $M_{sl}$ | Right and left spring torque value |

S5: During this phase, the metabolic rate is higher than that of walking without assistance, and the motor works. In this state, the left leg supports the swing of the right leg ($P_r = 0$, $P_l > 0$), and the hip angular velocity of the swinging leg is positive ($\omega_{hr} > 0$, $\omega_{hl} < 0$).

S6: During this phase, the metabolic rate is lower than when walking without assistance, and the motor does not work. In this state, the left leg supports the swing of the right leg ($P_r = 0$, $P_l > 0$), and the hip angular velocity of the swinging leg is negative ($\omega_{hr} < 0$, $\omega_{hl} < 0$).

The states that require the motor to work are S2 and S5. The meaning of the symbolic parameters used in the article is summarized in Table 2. The parameter expressions for each state are summarized in Table 3.

## State transition conditions

Based on the ground reaction forces (Fig 6(d)), joint angles (Fig 6(b)), and angular velocities (Fig 6(c)) in the data, the differences between each state are compared and the transition conditions between states are determined. The events that change the 7 states are as follows:

C0(S6 ⇒ S1): from right swing to double support, the transition condition between the two states is that the pressure of the right foot is greater than 0.

C1(S1 ⇒ S2): from double support to left swing, when the pressure on the left foot is 0, the state switches, and the motor starts to work when the switching event is triggered.

C2(S2 ⇒ S3): the angular velocity of the hip joint of the swinging leg (left leg) changes from a positive value to a negative value, when the angular velocity of the hip joint of the swinging leg (left leg) becomes 0, the state transitions and the motor stops.

C3(S3 ⇒ S4): from left swing to double support, the transition condition between the two states is that the pressure of the left foot is greater than 0.

**Table 3. Each state and its parameters.**

| States | Key parameters | | | |
|---|---|---|---|---|
| S1(About 0~20%) | $P_r > 0$ | $P_l > 0$ | $\theta_{hr} > \theta_{hl}$ | |
| S2(About 20~40%) | $P_r > 0$ | $P_l = 0$ | $\omega_{hr} < 0$ | $\omega_{hl} > 0$ |
| S3(About 40~50%) | $P_r > 0$ | $P_l = 0$ | $\omega_{hr} < 0$ | $\omega_{hl} < 0$ |
| S4 (About 50~70%) | $P_r > 0$ | $P_l > 0$ | $\theta_{hr} < \theta_{hl}$ | |
| S5(About 70~90%) | $P_r = 0$ | $P_l > 0$ | $\omega_{hr} > 0$ | $\omega_{hl} < 0$ |
| S6(About 90~100%) | $P_r = 0$ | $P_l > 0$ | $\omega_{hr} < 0$ | $\omega_{hl} < 0$ |
| S0(stop) | $P_r > 0$ | $P_l > 0$ | $\omega_{hr} \approx \omega_{hl} \approx 0$ | $\omega_{ar} \approx \omega_{al} \approx 0$ |

**Table 4. Key parameters of state transition.**

| Transition Conditions | Threshold |
|---|---|
| C0(Near 0%) | $P_r > 0$ |
| C1(Near 20%) | $P_l = 0$ |
| C2(Near 40%) | $P_l = 0$, $\omega_{hl} \approx 0$ |
| C3(Near 50%) | $P_l > 0$ |
| C4(Near 70%) | $P_r = 0$ |
| C5(Near 90%) | $P_r = 0$, $\omega_{hr} \approx 0$ |
| C6(stop 1) | $P_r > 0$, $\omega_{hr} \approx \omega_{hl} \approx \omega_{ar} \approx \omega_{al} \approx 0$ |
| C7(stop 2) | $P_l > 0$, $\omega_{hr} \approx \omega_{hl} \approx \omega_{ar} \approx \omega_{al} \approx 0$ |

C4(S4 $\Rightarrow$ S5): from double support to right swing, when the reaction force of the right foot is 0, the state switches, and the motor starts to work when the switching event is triggered.

C5(S5 $\Rightarrow$ S6): the angular velocity of the hip joint of the swinging leg (right leg) changes from a positive value to a negative value, when the angular velocity of the hip joint of the swinging leg (right leg) becomes 0, the state transitions and the motor stops.

C6(S5 $\Rightarrow$ S0): from right swing to double support and stop, when the pressure of the right foot is greater than 0 and the angular velocity of each joint is approximately equal to 0, the state switches, and the motor stops when the switching event is triggered.

C7(S2 $\Rightarrow$ S0): from left swing to double support and stop, when the pressure of the left foot is greater than 0 and the angular velocity of each joint is approximately equal to 0, the state switches, and the motor stops when the switching event is triggered.

Thus the conditions that trigger the motor to start are C1 and C4. The conditions that trigger the motor to stop are C2, C5, C6, and C7. All state transition events are shown in Table 4.

## Motor control

The motor control of the active drive part of the hybrid-actuated lower extremity exoskeleton mainly involves three questions: when does it work? What are the trigger conditions for motor start and stop? How does the motor output torque when it working? The time period during which the motor operates was described in Section 4.1. The trigger conditions for motor start and stop were described in Section 4.2. Therefore, the torque output by the motor during operation needs to be determined.

As shown in Fig 9, the control strategy first detects the key parameters of the wearer's kinematics and dynamics to determine which state the person belongs to at the moment, and then detects the event of the state transition. The state transition occurs when the condition is satisfied. The motor starts when entering a certain state (S2 or S5). The tension sensor measures the force generated by the auxiliary spring. The torque generated by the spring at the ankle joint is calculated based on the mathematical model of the ankle joint assist scheme. The motor uses this torque value as a reference to output compensation torque to the ankle exoskeleton.

## Simulation experimental validation of control strategy

In order to verify whether the control strategy in this paper can achieve the desired effect: reduce the metabolic rate during the metabolic rising phase, thereby further reducing the overall metabolic consumption. The data of a weight-bearing walking experiment of one of the

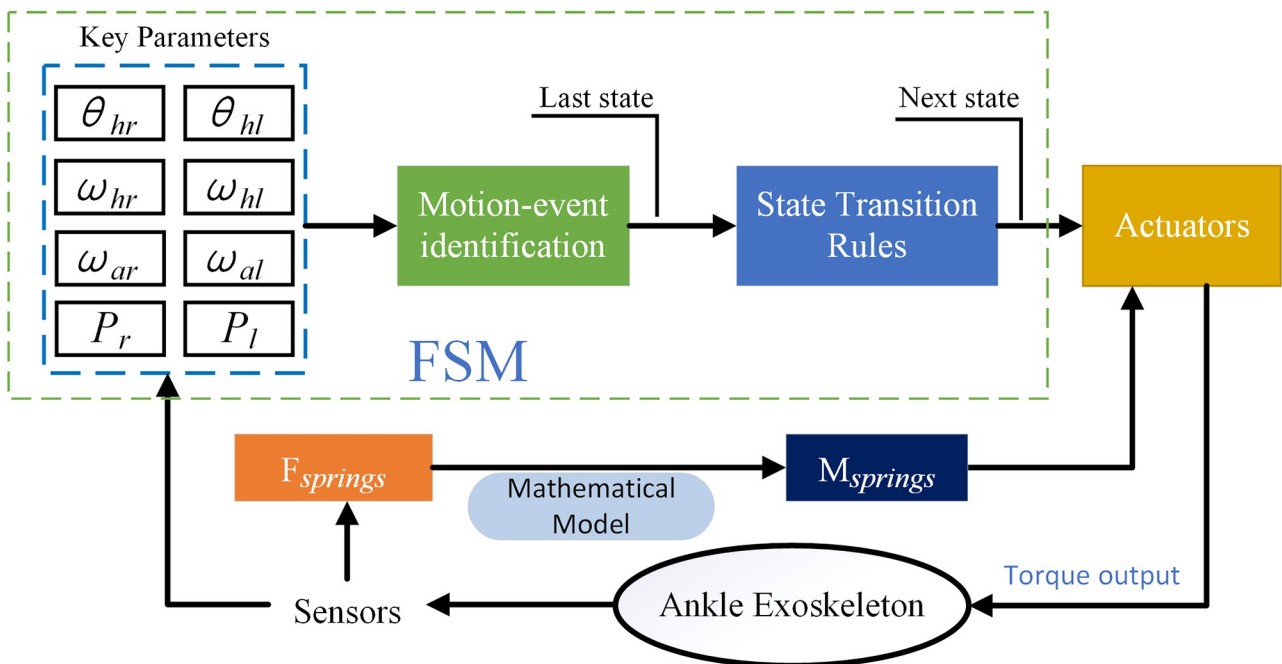

**Fig 9. The motor control of hybrid-actuated lower limb exoskeleton.** The dashed box is a finite state machine, which is responsible for the wearer's state recognition and state transition. Outside the dashed box is the sensing system and the drive system, which are responsible for controlling the motor to output an appropriate torque.

subjects was found from the data set for simulation verification, and the passive auxiliary spring and the active motor were added to the ankle joint at the same time. The active actuator applies a joint dorsiflexion moment at the ankle joint according to the method in Section 4.3.

Metabolic rate profiles calculated by adding a metabolic probe in OpenSim and running the CMC tool versus time are shown in Fig 10(a). The blue line in the figure is the case without assistance, the red line is the case where only the spring acts, and the green line is the case where the actuator is added based on the spring assistance and acts simultaneously. The gray area is the action stage of the active actuator, during which the introduction of the active

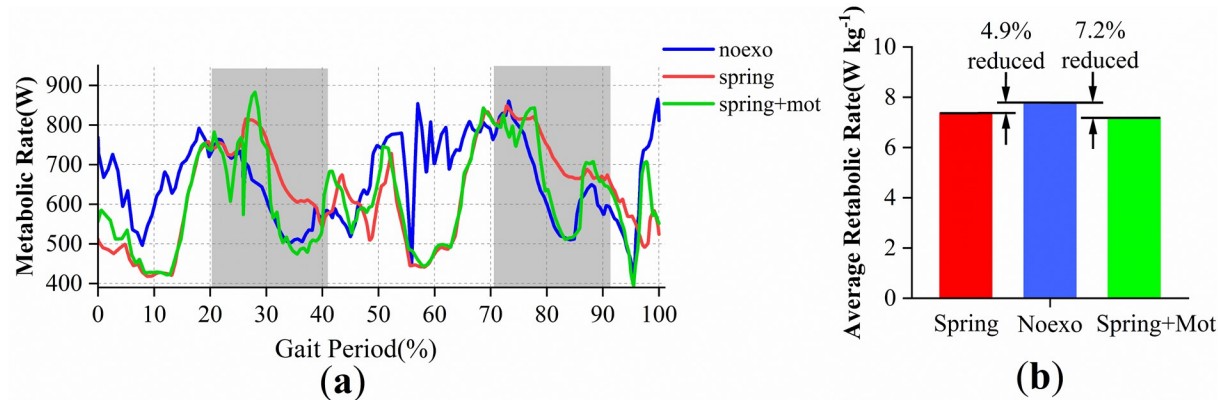

**Fig 10. Active and passive ankle exoskeleton simulation experiment results.** (a) Relationship between metabolic rate curve and time; (b) Average metabolic rate of different assistive methods.

actuator reduces the metabolic rate to a certain extent, while the white area is the stop-working stage of the actuator, at which time the red line and the green line coincide.

As shown in Fig 10(b), the average metabolic rate decreased by about 4.9% in the case of only the spring action, and the average metabolism decreased by about 7.2% after the introduction of the active actuator by comparing the average metabolic rate during the gait cycle.

## Discussion

We can find that the simulation results of the three passive-assist regimens in different subjects indicated that the ankle-assist regimen presented the greatest metabolic reduction in weight-bearing walking. This result is similar to the effect of most current experimental ankle plantar-flexion devices [27–30]. However, it also be found that the hip assist scheme differs from most current experimental hip flexion devices [4,31–33]. The major reason is that the metabolic base of weight-bearing walking is greater than that of natural walking.

In this paper, we choose the data of weight-bearing walking for simulation. The ideal devices without weights were added to the musculoskeletal model. Considering that the power and control system of the hybrid-actuated lower limb exoskeleton and its weight will load a large weight on the wearer. A load needs to be added to the model and simulated using the weighted walking data to bring the simulation closer to reality.

In addition, simulation experiments have limitations. Subject kinematics in the dataset did not change adaptively with the addition of assistive devices. Some studies have shown that people's adaptive changes after wearing assistive devices are a consideration in the experimental testing of devices [34–36]. There are errors in the metabolic values obtained from our simulations. However, this study focused on the changes in metabolic rate under different adjuvant regimens, so this effect can be ignored for our research goals.

## Conclusions

This paper proposed a conceptual design approach for a hybrid-actuated lower extremity exoskeleton based on energy consumption simulation. Metabolic expenditure during weight-bearing walking for 3 preset lower extremity assistance regimens was assessed using Open-Sim simulations. The passive energy storage structure of the ankle joint that can minimize metabolic expenditure was found by comparing the simulation results of 3 adjuvant regimens in 7 subjects. Therefore, a novel hybrid-actuated cooperative control strategy is proposed based on the finite state machine and the metabolic rate changes of human weight-bearing walking. The simulation results show that the control strategy further reduces the metabolic rate of the human body when walking with weight. Finally, the effectiveness of the control strategy and the feasibility of the conceptual design method of the hybrid-driven lower extremity exoskeleton is verified. In the future, more metabolic experiments of walking without loading should be proceeded to optimize the design of the hyper-actuated lower-limb exoskeleton.

## Author Contributions

**Conceptualization:** Qiaoling Meng.

**Data curation:** Bolei Kong.

**Formal analysis:** Bolei Kong.

**Funding acquisition:** Hongliu Yu.

**Methodology:** Qiaoling Meng.

**Project administration:** Qiaoling Meng.

**Software:** Bolei Kong.

**Supervision:** Hongliu Yu.

**Validation:** Bolei Kong, Qingxin Zeng, Cuizhi Fei.

**Writing – original draft:** Qiaoling Meng, Bolei Kong.

**Writing – review & editing:** Qiaoling Meng.

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
