## [Decision Letter · Decision Letter 0]

15 Feb 2023

PONE-D-22-32591Concept Design of Hybrid-Actuated Lower Limb Exoskeleton to Reduce the Metabolic Cost of Walking with Heavy LoadsPLOS ONE

Dear Dr. Yu,

Thank you for submitting your manuscript to PLOS ONE. After careful consideration, we feel that it has merit but does not fully meet PLOS ONE’s publication criteria as it currently stands. Therefore, we invite you to submit a revised version of the manuscript that addresses the points raised during the review process.

We look forward to receiving your revised manuscript.

Kind regards,

Zhan Li, PhD

Academic Editor

PLOS ONE

Journal Requirements:

Reviewers' comments:

Reviewer's Responses to Questions

**Comments to the Author**

1. Is the manuscript technically sound, and do the data support the conclusions?

Reviewer #1: Yes

Reviewer #2: Yes

2. Has the statistical analysis been performed appropriately and rigorously? 

Reviewer #1: Yes

Reviewer #2: Yes

3. Have the authors made all data underlying the findings in their manuscript fully available?

Reviewer #1: Yes

Reviewer #2: Yes

4. Is the manuscript presented in an intelligible fashion and written in standard English?

Reviewer #1: Yes

Reviewer #2: Yes

5. Review Comments to the Author

Reviewer #1: This paper proposed a design method for a hybrid-actuated lower limb exoskeleton to reduce the metabolic cost of walking. Three human-machine coupling models with passive springs were simulated and analyzed based on the method of simulation muscle driving. Then, this paper also proposed a control strategy based on the metabolic cost models for hybrid-actuated exoskeletons. The simulated experiments showed that the average metabolic cost decreased by 7.2% with both spring and motor. The work is interesting. The proposed method provides a new solution to improve lower limb exoskeletons.

Overall, the paper is well organized and its presentation is good. However, some minor issues still need to be improved.

1. The experiment results could be added in the abstract to obtain complete information for the readers.

2. The captions of figures in this paper are not very clear to the readers. The reviewer suggests that the authors could provide more information about the figures. For instance, Figure 2 (a),(b), and (c) meanings could be added in the caption.

Therefore, the reviewer suggests that the manuscript can be accepted after minor revision.

Reviewer #2: This paper proposes the conceptual design method for a hybrid-actuated lower limb exoskeleton based on energy consumption simulation. The work is interesting. The content is meaningful. Minor concerns includes:

(1) The quality of figures 1-2 should be improved.

(2) The abscissa of figure 5(a), 6 and 10(a) should be gait cycle instead of time.

6. PLOS authors have the option to publish the peer review history of their article (what does this mean?). If published, this will include your full peer review and any attached files.

Reviewer #1: No

Reviewer #2: No

---

## [Author Response · Author response to Decision Letter 0]

21 Feb 2023

Reviewer #1: This paper proposed a design method for a hybrid-actuated lower limb exoskeleton to reduce the metabolic cost of walking. Three human-machine coupling models with passive springs were simulated and analyzed based on the method of simulation muscle driving. Then, this paper also proposed a control strategy based on the metabolic cost models for hybrid-actuated exoskeletons. The simulated experiments showed that the average metabolic cost decreased by 7.2% with both spring and motor. The work is interesting. The proposed method provides a new solution to improve lower limb exoskeletons.

Overall, the paper is well organized and its presentation is good. However, some minor issues still need to be improved.

Comments:

1. The experiment results could be added in the abstract to obtain complete information for the readers.

Responses:

Thank you very much for your careful comments. We have added the experimental results in the abstract of the article. Please check out the file (b) that the specific changes with highlight.

Comments 

2. The captions of figures in this paper are not very clear to the readers. The reviewer suggests that the authors could provide more information about the figures. For instance, Figure 2 (a),(b), and (c) meanings could be added in the caption.

Responses:

Thank you very much for your careful comments. We have revised and supplemented the title of all the figures in the article. Please check out the file (b) that the specific changes of the figures title are highlighted.

Reviewer #2: This paper proposes the conceptual design method for a hybrid-actuated lower limb exoskeleton based on energy consumption simulation. The work is interesting. The content is meaningful. Minor concerns includes:

Comments:

(1) The quality of figures 1-2 should be improved.

Responses:

Thank you very much for your careful comments. We have optimized the definition of Figure 1 and Figure 2. Please check ‘Fig 1.tif’ and ‘Fig 2.tif’ in folder (d).

Comments:

(2) The abscissa of figure 5(a), 6 and 10(a) should be gait cycle instead of time.

Responses:

Thank you very much for your careful comments. We modify the abscissa in Figure 5 (a), Figure 6 and Figure 10 (a) to the gait period. Please check ‘Fig 5.tif’, ‘Fig 6.tif’and ‘Fig 10.tif’ in folder (d).

---

## [Editor Report · Decision Letter 1]

23 Feb 2023

Concept Design of Hybrid-Actuated Lower Limb Exoskeleton to Reduce the Metabolic Cost of Walking with Heavy Loads.

PONE-D-22-32591R1

Dear Dr. Yu,

We’re pleased to inform you that your manuscript has been judged scientifically suitable for publication and will be formally accepted for publication once it meets all outstanding technical requirements.

Kind regards,

Zhan Li, PhD

Academic Editor

PLOS ONE